# Effects of Demographic Variables on Subjective Neurocognitive Complaints Using the Neurocognitive Questionnaire (NCQ) in an Aged Japanese Population

**DOI:** 10.3390/ijerph16030421

**Published:** 2019-02-01

**Authors:** Michiko Yamada, Reid D. Landes, Ayumi Hida, Kayoko Ishihara, Kevin R. Krull

**Affiliations:** 1Departments of Clinical Studies and Statistics, Radiation Effects Research Foundation, 5-2 Hijiyama Park, Minami-ku, Hiroshima 732-0815, Japan; rdlandes@uams.edu (R.D.L.); ayumih@rerf.or.jp (A.H.); kishihara-kawase@umin.ac.jp (K.I.); 2Department of Biostatistics, University of Arkansas for Medical Sciences, Little Rock, AR 72205, USA; 3Department of Epidemiology and Cancer Control, St. Jude Children’s Hospital, Memphis, TN 38105-3678, USA; kevin.krull@stjude.org

**Keywords:** subjective neurocognitive complaint, aging, education, demographic factors, questionnaire, factor analysis

## Abstract

Objectives: In an aged Japanese population, we investigated associations of demographic variables with subjective neurocognitive complaints using the Neurocognitive Questionnaire (NCQ). Methods: Participants (*N* = 649) provided answers to the NCQ in both 2011 and 2013. Using fully-completed NCQs from 503 participants in 2011, we identified latent factors of subjective neurocognitive complaints using exploratory factor analysis; then examined associations of demographic variables with the identified factors for all 649 participants over the two years. We also examined changes in factor scores over the 2-year period. Results: We identified four factors representing 20 of the 25 NCQ items and labelled them metacognition, emotional regulation, motivation/organization, and processing speed. In a regression model using all participants, we observed linear deterioration with age on emotional regulation and linear-quadratic deterioration with age on the other factors. Less education was associated with more problems for all factors, but we detected no evidence of interaction between age and education. In 314 participants completing both assessments, paired *t*-tests comparing the 2013 to 2011 responses corroborated the regression results, except for emotional regulation. Conclusions: On the NCQ, older age and less education were associated with more subjective neurocognitive complaints. This is compatible with the association of the same factors with objective cognition and suggests that subjective cognitive complaints complement objective cognition as a prodrome of non-normative cognitive decline.

## 1. Introduction

In older persons, an increase in subjective cognitive complaints are related to a deterioration of simultaneously measured performances on neurocognitive tests [1]. Those with subjective cognitive complaints, even in the absence of current objective cognitive deficits, experience more objective cognitive decline over time [2,3]. These results suggest that subjective cognitive complaints may be a prodrome of non-normative cognitive decline. Although a questionnaire survey on subjective cognitive complaints is a simple and cost-effective method to examine neurocognitive function, few instruments that evaluate subjective cognition are used globally, and there is a dearth of reports on subjective cognition, except for subjective memory complaints [4]. In aging research, subjective cognitive complaints have received increased attention in recent years [5]. Some studies report inconsistencies between subjective and objective memory [6] and possible clinical and practical implications related to subjective cognitive complaints [7]. Self-report measures currently used to assess subjective cognitive decline exhibit wide variation in response options and questionnaire contents, and little is known about key features, such as the interplay of demographic factors and subjective cognition [5]. 

The adult version of the Behavior Rating Inventory of Executive Function (BRIEF-A) questionnaire evaluates subjective cognitive complaints [8] and has been used globally, including in studies of adults with attention-deficit hyperactivity disorder in Japan, Korea, and Taiwan [9]. The Neurocognitive Questionnaire (NCQ), a subjective self-reported measure of neurocognitive function, was developed to investigate the neurocognitive effects of therapy, including radiotherapy, for the Childhood Cancer Survivors Survey (CCSS) [10]. The NCQ consists of a subset of items developed in conjunction with the BRIEF-A that includes a behavioral regulations factor and a metacognition factor [11,12], and additional items specific to processing speed, memory, and academic functioning. In the CCSS, the NCQ was administered to child cancer survivors and their cancer-free siblings, and four factors (Task Efficiency, Emotional Regulation, Organization, and Memory) were found based on 19 of 25 items [10]. The NCQ was evaluated as a practical and efficient tool for monitoring neurocognitive outcomes with excellent reliability and validity in adult survivors of pediatric cancer [10]. Although age-related differences in cognitive function across various domains exist, [13] the use of the NCQ in older persons or Asians has not been reported.

Since 1958, the Adult Health Study (AHS) of the Radiation Effects Research Foundation (RERF) has conducted biennial health check-ups of people who were in Hiroshima or Nagasaki at the time of the atomic bombings to investigate the health effects of exposure to ionizing radiation and, since 1995, has conducted semiannual mail surveys to obtain additional health information. In 2011 and 2013, we used the NCQ, delivered by mail to this cohort, to examine late-life neurocognitive functions. Using exploratory factor analysis, we defined one or more “factors” of the NCQ as underlying latent constructs of subjective neurocognitive complaints among the cohort control participants. We evaluated how age, sex, city of residence, and education relate to subjective neurocognitive complaints by using factors defined by the exploratory factor analysis as dependent variables. We also examined changes of subjective neurocognitive complaints over 2 years to evaluate a participant’s longitudinal changes; those of objective cognition, however, might be masked by a ceiling effect [14] or a practice effect [15,16]. These examinations elucidate features of subjective cognition, which are little known compared with those of objective cognition. Subsequent analysis on NCQ responses for those exposed to radiation will later be conducted considering the model obtained for control participants (those who were not exposed to radiation).

## 2. Methods

### 2.1. Participants

We mailed the NCQ to participants during the last 6 months of both 2011 and 2013 through the AHS semiannual mail surveys. In the first administration, we sent the questionnaire to 2791 AHS subjects (atomic bomb survivors and their controls) who were aged 66 to 89 years, and 1819 (65.2%) of them mailed their replies back. To provide NCQ reference data for the aged Japanese, we limited analyses to the non-exposed (control) participants, who responded at a similar rate to their radiation-exposed counterparts. Among the 718 non-exposed respondents, 503 (70%) answered all NCQ questions at the 1^st^ administration, and the remaining 215 answered them partially (participants were not required to answer all the items). At the 2nd administration, 466 participants among 670 non-exposed respondents answered all NCQ questions, and 314 participants answered all NCQ questions at both administrations. The RERF Institutional Review Boards approved this study, and all participants provided informed consent.

### 2.2. Measurements

The NCQ is a self-administered screening tool for subjective neurocognitive complaints that are expressed in 25 statements. In our study, the participants received written instructions on how to report the degree to which they experienced any of the 25 problems over the past six months [10]. Possible responses were rated on the following scale: 1 = “Never a Problem,” 2 = “Sometimes a Problem,” and 3 = “Often a Problem.” Higher scores indicated poorer perceived neurocognitive function. 

### 2.3. Statistical Analysis

We performed an exploratory factor analysis on the first administration questionnaires, for which all 25 NCQ items were completed according to instructions (*N* = 503). We used a promax rotation and maximum likelihood estimation of factor loadings, considering the likelihood that the factors in any multifactor model would be correlated to some degree. Items with a factor loading of ≥0.40 defined the factor. We assessed internal consistency of the items within each factor with Cronbach’s (raw) alpha. We computed total NCQ scores as the mean of all 25 items; thus, total NCQ scores ranged from 1 to 3. Likewise, we computed factor scores as the mean of all items making up the factor, and scores ranged from 1 to 3. We also calculated Pearson’s correlations among the factors, which, computed pairwise, provided insight into any overlap of information provided by a pair of factors. 

For the following analyses, we report results on two samples—All Available (AA) and Complete Case (CC). Those who provided at least one factor score at either administration made up the AA sample. Those who answered all 25 NCQ items at both administrations made up the CC sample. 

We aimed to estimate change in the NCQ and each of the factors due to aging while controlling for sex, education, and city using the AA sample to maximize sample size. To analyze the scores, we used linear mixed models, which allowed us to (i) account for repeatedly measured scores from the 2 administrations and (ii) use the one-time scores from those participants who provided a score at only one of the administrations. Specifically, we accounted for within-participant correlation with a compound symmetric (a.k.a. exchangeable) structure, which assumes the score variance is constant at both administrations and that observations within a participant are correlated (this is a repeated measures analysis). For each score, we first built an age (at administration) model, choosing between fitting a purely linear age effect or a linear-quadratic age effect. Using the chosen age model, we individually tested effects of sex, education, and city at the 0.10 *p*-level. We then tested whether a significant cofactor interacted with each of the other cofactors. We also tested whether the linear age effect depended on the level of a significant cofactor. Finally, we used the model that had either all significant main effects or all significant main effects and main effects included in a significant interaction. Estimated effects are presented with their 95% confidence intervals. Separate from the aging models adjusted for cofactors, we provide summary statistics on the differences between scores from the 1st and 2nd administrations among the CC sample for evaluation of individual longitudinal changes; these are equivalent to paired *t*-tests. All analyses were conducted using SAS/STAT software, version 9.4, SAS System for Windows (SAS Institute, Cary, NC, USA). 

## 3. Results

In the first administration, 503 participants (mean age ± SD, 73.3 ± 5.8 years) answered all 25 NCQ questions; we used those responses for the exploratory factor analysis. Table 1 displays the factor loadings on the NCQ items, mean and standard deviation of score, and internal consistency measures (Cronbach’s alpha) for each factor. The mean (SD) for the total NCQ score was 1.33 (0.37). We obtained 4 factors representing 20 of the 25 items and categorized them as metacognition (Factor 1, 9 items), emotional regulation (Factor 2, 5 items), motivation/organization (Factor 3, 4 items), and processing speed (Factor 4, 2 items). Table 1 shows the 4 factors we identified using the NCQ as well as items represented for a 2-factor solution (behavioral regulation and metacognition) based on the BRIEF-A [12] and a 4-factor solution (task efficiency, emotional regulation, organization, and memory skills) based on the NCQ in the CCSS [10]. Items identified as a metacognition factor based on the BRIEF-A loaded on the metacognition factor and motivation/organization factor. The items identified as a task efficiency factor based on the NCQ in the CCSS loaded on the metacognition, emotional regulation, motivation/organization, and processing speed factors. Since recent literature using the BRIEF-A reported a better fit for a three-factor solution [17], where the behavioral regulation factor split into a behavioral regulation factor and an emotional regulation factor, the emotional regulation factor in this study (Table 1) was comparable to the behavioral regulation factor in the BRIEF-A or the emotional regulation factor based on the NCQ in the CCSS. We found processing speed to be an independently identified factor. The differences of the identified factors between CCSS and our study may be due to differences in age-related changes of executive function domains since the former involved the middle aged, and the latter the elderly. Table 2 provides the Pearson’s correlations for each factor pair.

After determining the 4 factors, we computed factor scores for the 649 participants in the AA sample. Table 3 displays the numbers of the AA sample providing scores for the NCQ and each of the 4 factors at (i) both administrations, (ii) only the 1st, and (iii) only the 2nd. 

Table 4 provides participant characteristics for the sample used in the exploratory factor analysis (EFA), the AA, and the CC samples. Because the CC sample was wholly contained in the EFA sample, and the EFA wholly contained in the AA sample, we did not statistically compare these samples, but do make the following notes. The AA sample had proportionately more women than the CC sample, and proportionately fewer participants with a university-level education (more than 15 years of education). The EFA sample had proportionately fewer Hiroshima participants than the AA and CC samples. The EFA sample values for the other characteristics ranged between those for the AA and the CC samples. Only one mildly demented participant was included in each administration.

In the mixed model based on the AA sample, chosen models for each score and estimates for the terms in the models appear in Table 5. For the NCQ and all 4 factors, subjective cognition deteriorated (scores increased) linearly with increasing age. For Factors 1, 3, and 4, (metacognition, motivation/organization, and processing speed) though, the *rate* of deterioration increased, with age showing a linear-quadratic effect. Those with less formal education had higher scores on all of the scores than those with more education. This was particularly true for Factors 1 and 2 (metacognition and emotional regulation) (*p* < 0.05 in both the univariate and multivariate models). For the total NCQ and Factors 3 and 4 (motivation/organization and processing speed), evidence was not as strong (*p* < 0.05 in the univariate model and 0.05 < *p* < 0.10 in the multivariate model). For all scores, there was no evidence that increased scores with age depended on education (all age × education interactions had *p* > 0.282, results not shown). Only for Factors 3 and 4 (motivation/organization and processing speed) did scores show some evidence that women and men might differ. For both scores, women had higher scores than men. It is well known that for those born before the end of World War II, men tended to stay in school longer than women. But the possible sex differences in Factors 3 and 4 did not depend on education level (both sex × education interactions had *p* > 0.468, results not shown). Additionally, we found no evidence that the age-related increases in all 5 scores differed between men and women (*p* > 0.376 for all age × sex interactions, results not shown). No factors showed evidence of city difference. Figure 1 and Figure 2 display the estimated effect of age on total NCQ score and Factor 1–4 scores at the mean education level. 

CC sample scores from the 1st to the 2nd administration indicate significant worsening of total NCQ, metacognition, motivation/organization, and processing speed factors, but not emotional regulation, for all age categories (Table 6). Greater score differences were observed in older participants (aged 80 years or more) than in younger participants for total NCQ and factors, except for emotional regulation. 

## 4. Discussion

Most cognitive dysfunction literature is devoted to objective cognitive dysfunction, and articles about subjective cognitive dysfunction are limited. Many instruments can evaluate objective cognition, including the Mini-Mental State Examination (MMSE) [18] and the Alzheimer’s Disease Assessment Scale [19]. Although many articles discuss subjective memory complaints, considerable debate surrounds the relationship between subjective memory complaints and objective memory, with some studies reporting significant effects and others reporting null results [4,6]. In a meta-analysis, Crumley and colleagues found that the association between subjective and objective memory was small, and age, years of education, sex, depression symptoms, and length and format of subjective memory measures were correlated significantly with its size [6]. Inconsistency between subjective and objective memory might be due to lack of reliability of the self-report instrument, methodological issues such as a non-standardized definition of subjective memory complaints and varied reference periods, and a small number of study subjects [4,5]. Regarding subjective cognition other than subjective memory complaints, the heterogeneity of approaches to subjective cognitive decline and lack of knowledge about the key features of subjective cognition have also been noted [5] based on the Subjective Cognitive Decline Initiative Working Group [20]. Although a definition of subjective cognitive complaints has not been operationalized [5,21], we investigated the identified factors of the NCQ to elucidate that feature. 

Regarding the effect of demographic variables on subjective cognitive complaints, Jonker and colleagues reported a positive correlation of memory complaints with age and a negative correlation with years of education, but no significant correlation with sex [1]. Snitz and colleagues reported similar correlations between demographic variables and overall subjective cognitive complaints, including 24 items examined in four community studies in the United States [22]. In the LIFE-Adult–Study, on the other hand, the prevalence rate of subjective memory complaints did not vary significantly with age, sex, or education [7]. Little is known about effects of demographic variables on individual subjective cognitive domains other than memory. There are more detailed reports on the association of demographic variables with objective cognitive function including for overall objective cognition [23,24], individual objective cognitive domains [13,25,26], and for interactions between demographic variables [24,27,28]. 

The accelerated deterioration with age of metacognition, motivation/ organization, and processing speed factors in our study resembled the increased deterioration with age in overall objective cognitive function in instruments such as MMSE [23] and the Cognitive Ability Screening Instrument (CASI) [29]. Although the subjective complaints could be caused by different mechanisms (psychological and/or biological) other than objective impairment, the underling mechanism for subjective complaints might have affected substantial cognitive impairment by subsequent interactive changes of psychological and biological status. Our finding of predominant accelerated deterioration with age in the very old for the processing speed factor was similar to findings for objective measures of processing speed in the Rotterdam study [13]. In our study, we also observed significant within-person deterioration over a 2-year period in metacognition, motivation/organization, and processing speed factors (Table 6). In objective cognitive function, the ceiling effect [14] and practice effect induced by repetition of the test masked the longitudinal effect of age, especially in early follow-up [15,16].

Although we observed no significant sex effects in our factors, women tended to suffer a little more than men in motivation/organization and processing speed. Sex differences in objective cognition have shown inconsistent results for various objective cognitive domains [25,26]. Hayat and colleagues reported no sex difference for global cognition, but poorer performance in men for memory and attention [25]. Rextoth and colleagues reported better memory performance in women, better reasoning performance in men, but no sex difference for processing composite [26]. Our study results are compatible with previous reports that increased education is independently associated with better performance in objective cognitive domains [16,23,24]. We did not find age effects varied with education (i.e., no age × education interaction). A previous report on AHS participants born before September 1932 found that objective cognition (evaluated with the CASI) decreased with increasing age and fewer years of formal education, but it, too, failed to find a significant age × education interaction [29]. Regarding the interactive effects of age and education on objective cognition, Kirton and Dotson reported that age differences in executive function were reduced at higher education levels [27], but no interaction was reported on MMSE [24] and individual cognitive domains [30]. 

The strength of this study is its relatively large sample of an aged population and its longitudinal information. A limitation is the large proportion of participants who did not answer all the NCQ questions. A higher proportion of older women failed to complete the questionnaire, but we did not investigate the reason. Additionally, we should consider the limitations to quantifying an inherently subjective phenomenon such as a complaint [5].

## 5. Conclusions

In conclusion, we evaluated subjective neurocognitive complaints among aged Japanese using the NCQ. Our exploratory factor analysis defined four factors—metacognition, emotional regulation, motivation/organization, and processing speed. Older age and lower education levels were associated with more subjective neurocognitive complaints. Although those demographic associations with subjective neurocognitive complaints are compatible with those for objective cognition, this result suggests that subjective cognitive complaints complement objective cognition as a prodrome of non-normative cognitive decline. To explore this possibility, further investigation with different populations and instruments, and comparisons with objective cognition, are required. 

## Figures and Tables

**Figure 1 ijerph-16-00421-f001:**
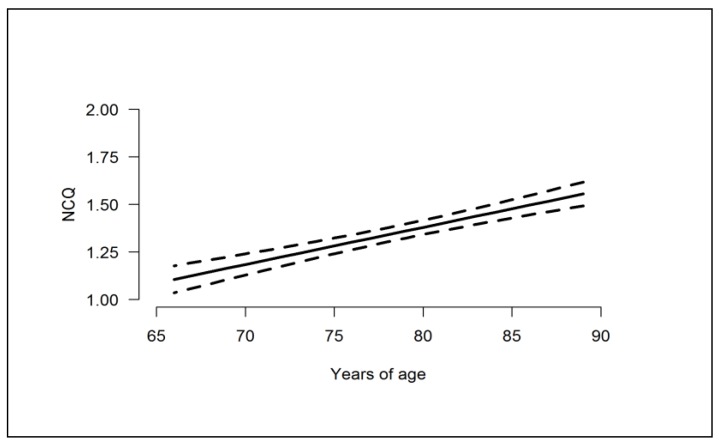
Estimated effect of age and its 95% confidence interval on NCQ score.

**Figure 2 ijerph-16-00421-f002:**
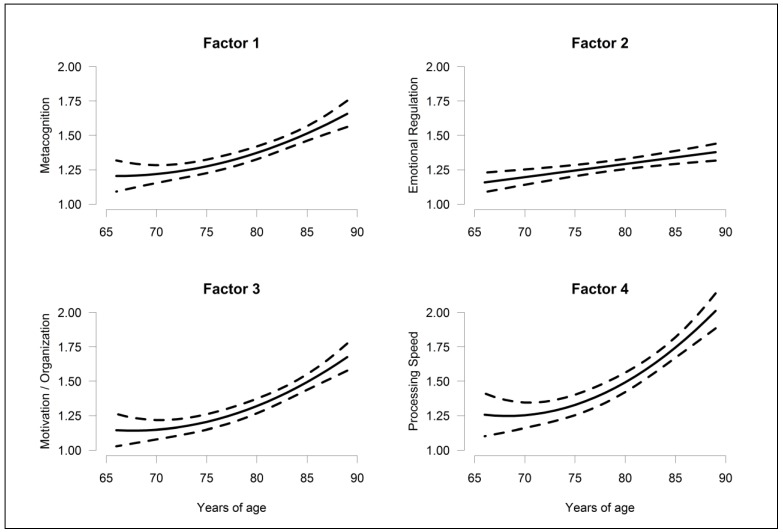
Estimated effects of age and their 95% confidence interval on Factors 1–4 scores.

**Table 1 ijerph-16-00421-t001:** Factor loadings, summary statistics of factor scores, and internal consistency measures (*N* = 503).

#	NCQ Item	Factor 1	Factor 2	Factor 3	Factor 4	^a^ BRIEF-A	^a^ CCSS NCQ
24	Trouble solving math problems	**0.78**	0.08	−0.05	0.01		-
20	Trouble remembering things	**0.68**	0.04	0.06	0.01	metacognition	memory
22	Reads slowly	**0.55**	0.03	−0.02	0.19		task
21	Trouble prioritizing activities	**0.51**	−0.02	0.40	−0.08	metacognition	task
17	Trouble with multitasking	**0.47**	−0.03	0.24	0.23	metacognition	task
25	Does not work well under pressure	**0.44**	0.29	−0.03	0.22		task
10	Different ways to solve a problem	**0.44**	0.16	0.29	0.04		-
13	Forgot what doing in middle of things	**0.42**	0.05	0.28	−0.02	metacognition	memory
5	Forgets instructions easily	**0.40**	0.09	0.28	0.06	metacognition	memory
1	Gets upset easily	0.01	**0.74**	−0.12	0.12		emotional
9	Mood changes frequently	0.02	**0.72**	0.24	−0.07	behavioral	emotional
8	Gets frustrated easily	0.16	**0.70**	0.02	−0.10		emotional
16	Easily overwhelmed	0.16	**0.50**	0.01	0.19		task
11	Impulsive	−0.10	**0.46**	0.36	0.02	behavioral	-
6	Problems completing my work	0.00	0.04	**0.67**	0.21	metacognition	task
4	Disorganized	0.03	0.03	**0.48**	0.27	metacognition	organization
14	Problems with self-motivation	0.35	−0.01	**0.46**	0.11	behavioral	task
12	Trouble finding things in bedroom	0.29	0.04	**0.40**	0.15	metacognition	organization
23	Slower than others	0.19	−0.07	0.13	**0.69**		task
2	Takes longer to complete work	−0.06	0.06	0.27	**0.62**		task
3	Does not think of consequences	0.12	0.22	0.31	0.09	behavioral	-
7	Difficulty recalling things learned before	0.35	0.18	0.12	0.04		memory
15	Underachiever	0.36	0.14	0.09	0.26		-
18	Blurts things out	0.32	0.27	0.08	−0.08		-
19	Desk/workspace a mess	0.11	0.04	0.29	0.23		organization
	Mean factor score	1.34	1.26	1.25	1.38		
	Standard deviation of factor score	0.42	0.38	0.42	0.55		
	Cronbach’s	0.911	0.851	0.842	0.817		

Factor 1: metacognition, Factor 2: emotional regulation, Factor 3: motivation/organization, Factor 4: processing speed. Items 3, 7, 15, 18, and 19 did not meet criteria to load on a factor; summing their scores, the mean (SD) was 1.41 (0.37) and Cronbach’s α was 0.731. ^a^ Factors reported in [10]: behavioral regulation (behavioral), task efficiency (task), memory skills (memory), organization skills (organization), emotional regulation (emotional). NCQ: The Neurocognitive Questionnaire, CCSS: the Childhood Cancer Survivors Survey, BRIEF-A: The adult version of the Behavior Rating Inventory of Executive Function. Items with a factor loading of ≥0.40 (shown in bold number) defined the factor.

**Table 2 ijerph-16-00421-t002:** Pearson’s correlations between each pair of factors. All six *p* < 0.0001 (*N* = 503).

Score	Correlations with		
	Factor 2	Factor 3	Factor 4
Factor 1	0.708	0.798	0.707
Factor 2	--	0.641	0.533
Factor 3	--	--	0.728

**Table 3 ijerph-16-00421-t003:** Sample sizes for each score type provided from the All Available sample (*N* = 649).

Administration	NCQ	Factor 1	Factor 2	Factor 3	Factor 4
Total	555	614	612	626	630
Only 1st administration	189 ^b^	177	168	165	151
Only 2nd administration	52	29	18	23	13
Both administrations	314 ^a,b^	408	426	438	466

^a^ These 314 made up the Complete Case (CC) sample. ^b^ These 503 made up the sample on which the exploratory factor analysis was performed.

**Table 4 ijerph-16-00421-t004:** Participant characteristics for the Complete Case (CC), Exploratory Factor Analysis (EFA), and All Available (AA) samples. Note: CC is wholly contained in EFA, and EFA wholly contained in AA.

Sample	*N*	City (%)		Sex (%)	^a^ Education (%)				^b^ Age
		Hiroshima	Nagasaki	Female	Primary School	High School	Junior University	University	Mean ± *SD*
CC	314	56.0	44.0	52.9	20.1	60.4	6.8	12.7	76.2 ± 5.8
EFA	503	53.7	46.3	58.7	22.8	60.5	6.9	9.8	77.3 ± 5.8
AA	649	55.2	44.8	61.6	23.7	61.0	7.3	8.0	77.6 ± 5.9

^a^ Sample size reduced by 6 for CC, 12 for EFA, and 16 for AA samples because of missing education information. ^b^ Age at first administration of the questionnaire.

**Table 5 ijerph-16-00421-t005:** Final models chosen for each score, with estimates (and 95% confidence intervals) for the terms in the model; “--” indicates term was not in the final model. Final models were chosen using the All Available (AA) Sample.

^a^ Score	^b^ Age-Linear	^b^ Age-Quadratic	^c^ Male	^d^ Education		
				HS vs. P	JU vs. P	U vs. P
NCQ	0.195; (0.147, 0.243)	--	--	−0.065; (−0.138, 0.008)	−0.152; (−0.280, 0.023)	−0.111; (−0.226, 0.005)
Factor 1	0.199; (0.146, 0.251)	0.085; (0.008, 0.161)	--	−0.104; (−0.183, −0.026)	−0.180; (−0.318, −0.042)	−0.176; (−0.305, −0.048)
Factor 2	0.096; (0.048, 0.143)	--	--	−0.092; (−0.164, −0.020)	−0.208; (−0.330, −0.085)	−0.149; (−0.266, −0.033)
Factor 3	0.233; (0.180, 0.286)	0.118; (0.038, 0.198)	−0.062; (−0.130, 0.006)	−0.088; (−0.165, −0.010)	−0.130; (−0.265, 0.005)	−0.106; (−0.238, 0.026)
Factor 4	0.332; (0.262, 0.402)	0.178; (0.074, 0.282)	−0.089; (−0.181, 0.003)	−0.118; (−0.220, −0.015)	−0.067; (−0.248, 0.114)	−0.189; (−0.364, −0.013)

^a^ See Table 3 for sample sizes for each score. ^b^ Age was centered at the mean age of 77.6 year, and divided by 10; hence, regression slopes are the expected change in the score for a 10 year increase in age. ^c^ Estimates are relative to females. ^d^ Education was reported as primary school (P), high school (HS), junior university (JU), and university (U). Estimates are differences from primary school. Factor 1 is metacognition; Factor 2, emotional regulation; Factor 3, motivation/organization; and Factor 4, processing speed.

**Table 6 ijerph-16-00421-t006:** Correlation and differences in scores (95% confidence intervals) from the 1st to the 2nd administrations for the CC sample (*N* = 314).

Score	Correlation	Differences in Scores by Age Group					
		All CC	[65, 70)	[70, 75)	[75, 80)	[80, 85)	[85, 90)
NCQ	0.81	0.05; (0.02, 0.07)	0.04; (−0.02, 0.09)	0.05; (0.01, 0.10)	−0.01; (−0.06, 0.05)	0.07; (0.02, 0.12)	0.08; (−0.02, 0.18)
Factor 1	0.77	0.05; (0.02, 0.08)	0.03; (−0.03, 0.09)	0.07; (0.02, 0.13)	−0.01; (−0.07, 0.05)	0.08; (0.01, 0.15)	0.07; (−0.07, 0.21)
Factor 2	0.73	0.02; (−0.01, 0.05)	0.06; (−0.00, 0.12)	0.04; (−0.02, 0.10)	−0.05; (−0.11, 0.02)	0.01; (−0.05, 0.07)	0.05; (−0.04, 0.14)
Factor 3	0.73	0.06; (0.02, 0.09)	0.05; (−0.01, 0.11)	0.06; (0.01, 0.11)	0.00; (−0.07, 0.08)	0.07; (0.00, 0.13)	0.18; (−0.04, 0.39)
Factor 4	0.70	0.11; (0.06, 0.15)	0.01; (−0.07, 0.09)	0.06; (−0.02, 0.13)	0.07; (−0.06, 0.19)	0.20; (0.11, 0.29)	0.25; (0.03, 0.47)

Factor 1 is metacognition; Factor 2, emotional regulation; Factor 3, motivation/organization; and Factor 4, processing speed.

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
