# Peer review of "Effects of Demographic Variables on Subjective Neurocognitive Complaints Using the Neurocognitive Questionnaire (NCQ) in an Aged Japanese Population"

_ijerph, 2019, doi:10.3390/ijerph16030421_

Round 1

Reviewer 1 Report

Research article by Yamada et al demonstrates how NCQ can be used to evaluate associations between demographic variables and subjective neurocognitive complaints. Although studies about subjective cognitive dysfunction are limited in research, the authors create a novel way into bridging the gap by using a meta-analytic/epidemiology approach. Novelty and significance of this manuscript is high and is of great interest/importance to any researcher who studies neurocognitive disorders, biostatistics, and epidemiology. Please check text for minor errors and past tense usage (ex. using an vs. a; has vs. have).

Author Response

Reviewer #1

Research article by Yamada et al demonstrates how NCQ can be used to evaluate associations between demographic variables and subjective neurocognitive complaints. Although studies about subjective cognitive dysfunction are limited in research, the authors create a novel way into bridging the gap by using a meta-analytic/epidemiology approach. Novelty and significance of this manuscript is high and is of great interest/importance to any researcher who studies neurocognitive disorders, biostatistics, and epidemiology. Please check text for minor errors and past tense usage (ex. using an vs. a; has vs. have).

Thank you for your comments. We are very pleased that you are interested in this article.

Reviewer 2 Report

General Comments: I commend the authors on their work conducting this study. The investigation of methods to assess subjective cognitive impairment in large populations is needed to help the field move forward to understanding the often-observed disconnect between subjective and objective reports of cognitive performance. I have offered several comments/suggestions below that I feel will add to the impact of this manuscript.

Abstract

It is stated that you found a linear deterioration with age on emotional regulation. However, you then say that you did not find significant deterioration for emotional regulation. This needs to be revised for consistency. If the linear deterioration mentioned first is non-significant, please make note of this.

Introduction:

The first paragraph does not make reference to any specific population. I assume you are referring to older adults, however this is never explicitly stated.

Methods

How were the NCQ’s returned to the researchers? Why was the questionnaire not distributed via email (electronic survey)?

Results

Table 1 legend – remove “the” from the factor titles (e.g., “the metacognition” to “metacognition”)_

Lines 146-149 you state that identifying the factor of “processing speed” may be due to age-related differences in executive function between the current sample and that of the CCSS. However, as processing speed is not typically looked at as falling under the “executive function” umbrella, I would recommend stating a more general age-related difference in “cognitive performance”.

Line 162 – fix formatting to include this in the table legend.

Table 3 – for readability, I recommend formatting the order of the rows to be “only 1st”, “only 2nd”, and “both”.

Line 175 “scores deteriorated (increased)” is potentially confusing. I recommend editing to be more direct – “subjective cognition deteriorated (scores increased)”

Line 177 “suffered more deterioration” – I recommend changing to something such as “had higher scores”.

The reporting of results surrounding the effect of education on deterioration is quite confusing. I’m not able to discern the observed effects based on what is written (there IS an effect and there ISN’T an effect). Perhaps more clarity about which FACTOR you are referring to with each effect would make this easier to follow.

I recommend removing the qualitative language (i.e., “suffer”) in the results and discussion. Instead, please report the observations (i.e., “higher score”).

Figure 2 – while this is not a necessary change, I believe this figure would be more impactful if you removed the “(A)”, “(B)”, “(C)”, “(D)” and replaced it with “Factor 1”, etc… And, label the Y-axes the title of the particular factor (i.e., “metacognition”). This would allow the figure to stand on its own without the legend.

Discussion

As with the introduction, you make very general statements about cognition research. For instance, the area of research on cancer-related cognitive impairment mainly contain subjective cognitive performance research. Therefore, it would be good to identify the specific population this study is concerned with.

229 – take “etc.” out. List all factors, or state that you’re providing examples of a few.

You state various good reasons to explain for the disparity between subjective and objective measures of cognition in the literature. However, it should be noted that subjective complaints could be caused by different mechanisms (psychological and/or biological) than objective impairment, and could be a prodrome (as stated in the introduction).

244 typo: “effect of demographics” – change to “effects of demographics”

275 – remove “(they were not required to)” as all research participation is voluntary.

Author Response

Reviewer #2

General Comments: I commend the authors on their work conducting this study. The investigation of methods to assess subjective cognitive impairment in large populations is needed to help the field move forward to understanding the often-observed disconnect between subjective and objective reports of cognitive performance. I have offered several comments/suggestions below that I feel will add to the impact of this manuscript.

Thank you very much for your valuable comments/suggestions. We tried to improve this article by considering your review.

Abstract

It is stated that you found a linear deterioration with age on emotional regulation. However, you then say that you did not find significant deterioration for emotional regulation. This needs to be revised for consistency. If the linear deterioration mentioned first is non-significant, please make note of this.

 The former mentioned the results of the linear mixed model for all available participants. (We say “regression” in the abstract; though we do not explicitly say it is a linear mixed model.) The latter mentioned paired t-test results among complete participants in both 2011 and 2013. We rewrote the Abstract to avoid confusions.

 Introduction:

The first paragraph does not make reference to any specific population. I assume you are referring to older adults, however this is never explicitly stated.

We are referring to older adults as you assumed. We now clearly state our focus on an aged population.

Methods

How were the NCQ’s returned to the researchers? Why was the questionnaire not distributed via email (electronic survey)?

The NCQ was conducted through AHS semiannual mail surveys; this is mentioned in the Introduction section. We now describe how the respondents replied – by mail.  These subjects are well-familiar with mailed questionnaires associated with the AHS; further, older persons tend to be less familiar with email communication than younger populations.

Results

Table 1 legend – remove “the” from the factor titles (e.g., “the metacognition” to “metacognition”)_

 We removed “the”.

Lines 146-149 you state that identifying the factor of “processing speed” may be due to age-related differences in executive function between the current sample and that of the CCSS. However, as processing speed is not typically looked at as falling under the “executive function” umbrella, I would recommend stating a more general age-related difference in “cognitive performance”.

 To avoid confusion by this statement, we mentioned about general age-related differenced in the Introduction section.

Line 162 – fix formatting to include this in the table legend.

 I am sorry I could not accomplish what you recommended.

Table 3 – for readability, I recommend formatting the order of the rows to be “only 1st”, “only 2nd”, and “both”.

 We changed the order.

Line 175 “scores deteriorated (increased)” is potentially confusing. I recommend editing to be more direct – “subjective cognition deteriorated (scores increased)”

 We changed the statement.

Line 177 “suffered more deterioration” – I recommend changing to something such as “had higher scores”.

 We changed the statement.

The reporting of results surrounding the effect of education on deterioration is quite confusing. I’m not able to discern the observed effects based on what is written (there IS an effect and there ISN’T an effect). Perhaps more clarity about which FACTOR you are referring to with each effect would make this easier to follow.

 We rewrote the second paragraph from the bottom to avoid confusion.

I recommend removing the qualitative language (i.e., “suffer”) in the results and discussion. Instead, please report the observations (i.e., “higher score”).

 We changed the related statements in the Results section.

Figure 2 – while this is not a necessary change, I believe this figure would be more impactful if you removed the “(A)”, “(B)”, “(C)”, “(D)” and replaced it with “Factor 1”, etc… And, label the Y-axes the title of the particular factor (i.e., “metacognition”). This would allow the figure to stand on its own without the legend.

We have made the suggested changes.

Discussion

As with the introduction, you make very general statements about cognition research. For instance, the area of research on cancer-related cognitive impairment mainly contain subjective cognitive performance research. Therefore, it would be good to identify the specific population this study is concerned with.

 Although we are not sure that the changes we’ve made in the Discussion addresses your suggestion, we have added information about subsequent analyses for the specific population (those exposed to radiation) in the Introduction section in response to editor comment.

229 – take “etc.” out. List all factors, or state that you’re providing examples of a few.

 We added other factors.

You state various good reasons to explain for the disparity between subjective and objective measures of cognition in the literature. However, it should be noted that subjective complaints could be caused by different mechanisms (psychological and/or biological) than objective impairment, and could be a prodrome (as stated in the introduction).

 Thank you for your suggestion. We added the statement in the second paragraph from the bottom in first page of the Discussion section.

244 typo: “effect of demographics” – change to “effects of demographics”

 We have made the change.

275 – remove “(they were not required to)” as all research participation is voluntary.

We have removed this statement.